# Effect of Watering down Environmental Regulation on Residents’ Health in China: A Quasi-Natural Experiment of Local Officials’ Promotion Motivation

**DOI:** 10.3390/ijerph192416770

**Published:** 2022-12-14

**Authors:** Xiaojia Chen, Yue Chen, Yuanfen Li, Wei Xu

**Affiliations:** 1School of Public Administration, Guangzhou University, Guangzhou 510000, China; 2College of Economics, Jinan University, Guangzhou 510632, China

**Keywords:** water pollution, water quality supervision, official promotion, resident health

## Abstract

Environmental performance is increasingly important in promoting officials, whose pursuit of promotions and related behavior may affect the health of residents in their jurisdictions. In this study, we spatially matched Chinese river water quality monitoring station data, enterprise pollution emission data, and resident health data and quantified how Chinese officials pursuing promotions based on environmental performance affected resident health using a regression discontinuity design and difference-in-difference with interaction terms design strategy. The results show that the upstream–downstream disparity of environmental governance and pollutant emissions affects the residents’ health, medical treatment behavior, and medical expenditure. Furthermore, we identified the causal relationship between official promotion and upstream–downstream disparity and estimated the marginal effect of promotion on residents’ health. The study suggests that local officials limit the pollution emissions of enterprises in the upstream river to achieve environmental performance and relax the pollution restrictions of firms in the downstream river to achieve economic performance, such that the health of residents near the river is differentially affected.

## 1. Introduction

Improving water quality is one of the major challenges faced by humanity in the twenty-first century, because more than one third of the global population does not have access to safe drinking water [1,2,3]. Water is an important natural resource used for drinking and other developmental purposes, and safe drinking water is necessary for human health worldwide. Based on the data of the World Health Organization (WHO), 80% of known diseases are waterborne [4]. Severe water pollution problems are common in developing countries. China’s rapid economic development and industrialization since the 1970s have led to a sharp deterioration in water quality [5]. In monitored areas of major rivers in China, only 28% of the water is suitable for drinking. Approximately one third of the water does not meet the minimum national environmental water quality standards, making these rivers unsuitable for irrigation [6,7,8]. Providing safe water is a challenging task [9]. It is widely accepted that developing an appropriate policy framework is key to combating water pollution [10].

To achieve high Gross Domestic Product (GDP) growth, the Chinese government did not set strict targets for emission reductions and water quality improvement in the 1990s. However, China is facing increasing environmental challenges, including the deterioration of river water quality. Since the early 2000s, China has implemented strict water pollution control measures. The water quality reading from water quality monitoring stations is included in the assessment indicators of local officials [11]. This policy aims to form an incentive-compatible contractual relationship between central and local governments so that local officials can achieve the environmental goals set by the central government because of the promotion incentive [12]. In other words, the policy’s purpose is to ensure that local government officials (called agents) with advanced access to information act in the interest of the central government (principal) such that both parties maximize utility. While this incentive system can encourage local governments to implement the central government’s policies, it can also create incentives to cheat, as highlighted by the principal–agent dilemma [13].

China’s central government has mobilized local officials to protect the environment and meet its pollution reduction targets. The central government assigns requirements to provinces to reduce pollution emissions, and provincial governors must sign personal responsibility documents detailing their pollution reduction plans and commitments with the central government. Governors pressure city and county officials to reduce emissions. Given the solid political incentive to promote officials, local governments have turned to polluters whose emissions directly cause degraded river water quality. Because data from water quality monitoring stations can only reflect the water quality upstream of rivers, local governments mainly focus on polluting enterprises located upstream of water quality monitoring stations [12,14]. However, few studies have investigated how the principal–agent dilemma affects residents’ health.

This study aimed to quantify how the promotion behavior of Chinese officials in pursuit of environmental performance affects residents’ health. In a centralized system, the central government often adopts a goal-oriented incentive system that links the promotion of officials to a specific performance indicator to ensure that local governments are implementing specific pollution control policies. However, if the central government does not adequately monitor the implementation of policies, local officials tend to work hard in areas that are easy to watch but do nothing else. In this study, the central government initially hoped to improve water quality by mobilizing promotional incentives for officials. However, the central government can supervise local governments only through the control of water quality data from river monitoring stations. Because the river flows from high to low altitudes, readings from water monitoring stations can only reflect pollution emissions from upstream firms, and local government officials have strong incentives to regulate upstream polluters. Consequently, companies upstream of the river are heavily regulated, while many downstream companies remain entirely unregulated, thus leading to regulatory enforcement that deviates from the original intention of the central policy.

## 2. Materials and Methods

### 2.1. Data Source and Variables

#### 2.1.1. Dependent Variables

We used the questionnaire data from the China Health and Nutrition Survey (CHNS) database to measure residents’ health as a dependent variable. The results of the CHNS was carried out by the Chinese Center for Disease Control and Prevention and the American University of North Carolina during the period 1989–2011 in nine provinces (Liaoning, Jiangsu, Shandong, Henan, Hubei, Hunan, Guangxi, Guizhou, and Heilongjiang; see Figure 1) were used to determine the social and economic status, residents’ health, dietary preferences, and nutritional status. Four sets of questionnaire data from the CHNS were used: (1) “been sick in the last four weeks?” (1 for sick and 0 for healthy), (2) “seek formal medical care in last four weeks” (1 for yes and 0 for no), (3) “visit a folk doctor in the last four weeks” (1 for yes and 0 for no), and “expenditure on illness”(Yuan).

#### 2.1.2. Interaction Variables

Interaction variables in this study include the promotion variable and the pollution variable. The promotion variable captures local officials’ promotion (1 for official promotion and 0 otherwise), which collected the biographical information of 5032 individuals from the websites of local governments in China.

The pollution variables include wastewater emissions, ammonia nitrogen emissions, and chemical oxygen demand, obtained from the National Bureau of Statistics, which contains China’s most comprehensive firm-level pollution emission data from 2000 to 2011.

#### 2.1.3. Driving Variable

The driving variable is the distance between the polluters and the water quality monitoring station, which is essential for determining the cutoff of the regression discontinuity design. The process steps were as follows. First, a script was written in the Python programming language to automatically collect polluters’ latitude and longitude data through the Application Program Interface (API) provided by Google Maps. Second, we used ArcGis 14 to compare the elevation of the areas on both sides of the monitoring station—the higher elevation was considered upstream, and the lower was considered downstream. Finally, the distance of firms within 10 km of the water quality monitoring station was calculated. A positive and negative number represented an upstream and downstream distance, respectively.

The sources of data used for distance calculation were as follows: (1) water quality monitoring stations in river basins in China, as shown in Figure 1, were obtained from the China Ministry of Ecology and Environment; (2) a Digital Elevation Model (DEM) was used, which determined whether areas were upstream or downstream, i.e., a remote sensing topographic map with a resolution of 12.5 m mapped by the Advanced Land Observing Satellite (ALOS); (3) a digital river map from OpenStreetMap.

### 2.2. Regression Design

An inappropriate model may result in unreliable estimates for the relationships between indicators and explanatory variables [15]. To identify the effects of the central government’s environmental target-based policy on residents’ health, a regression discontinuity design (RDD) was used to estimate causality between the upstream–downstream environmental governance disparity and residents’ health, with the mechanisms of promotion and pollution estimated by the difference-in-difference (DiD) model.

The upstream–downstream environmental governance disparity is the discontinuous characteristic of the RDD in this study, which exogenously determines whether an individual in the sample is the control group or the treatment group (Figure 2). The discontinuous variation in the residents’ health is caused by the treatment state captured by the distance between the polluter and the water quality monitoring station. Based on previous studies [16,17,18], we set the following RDD:(1)$$Sick_{it}=\alpha +\beta \times D_{it}+f\left({\mathrm{Distance}}_{it}\right)+\delta_{i}+\delta_{t}+\varepsilon_{it},$$
where subscript i indicates the individual and t indicates the year, $D_{it}$ is the state variable (1 for upstream and 0 for downstream), ${\mathrm{Distance}}_{it}$ is a running variable that measures the distance from the polluter to the water monitoring station, $f\left({\mathrm{distance}}_{it}\right)$ is a polynomial function that can be set as triangular, Epanechnikov or uniform by Stata 14 statistical software, $\delta_{i}$ and $\delta_{t}$ are the regional fixed effect and year fixed effect, respectively, and $\varepsilon_{it}$ is a random disturbance term. α is an intercept coefficient and β is a model coefficient that captures the treatment effect.

In this study, the DID model was used to further identify the mechanism, pollution emission, and official promotion, from the upstream–downstream disparity to residents’ health. We set up the following DID model with the interaction term:(2)$$Sick_{ijt}=\alpha_{1}Upstream_{jt}+\alpha_{2}Interaction_{jt}+\alpha_{3}Upstream_{jk}\times \;Interaction_{jt}+\theta_{j}+\theta_{t}+\varepsilon_{ijt},$$
where i represents the individual; j represents the region, and t represents the year; $Sick_{ijt}$ is the explained variable; $Upstream_{jt}=1$ indicates that the polluter was located upstream of the water quality monitoring station; $Interaction_{jt}$ is a variable that uses pollution emission data when testing the pollution mechanism or uses data on officials’ promotions when testing the promotion mechanism; $\theta_{i}$ and $\;\theta_{t\;}$ are the region- and time-fixed effects, respectively; $\varepsilon_{ijt}$ is a random disturbance term; $\alpha_{1}$, $\alpha_{2}$, $\alpha_{3},$ and $\alpha_{4}$ are the model coefficients; $\alpha_{1}$ and $\alpha_{2}$ are the main effects of the upstream variables and interaction variable, respectively; and $\alpha_{3}$ is the effect of the interaction of upstream variables and interaction variable. The derivation of Model (3) shows that the marginal effect is the sum of the main and interaction effects. The model designed in this section uses the Stata 14 software for regression, the results of which are reported in the next section.

## 3. Results

### 3.1. Statistical Analysis

Table 1 shows the descriptive statistics of the main variables. First, there are four dependent variables, among which Sick1 captures the probability of being sick, Sick2 and Sick3 are used for testing the medical treatment behavior of residents, and Sick4 identifies the cost of medical treatment. Second, the driving variable, distance, determines the treatment in the RDD. Third, the interaction variable is used for estimating the mechanism of the upstream–downstream disparity of environmental governance and residents’ health, which includes the pollution mechanism and promotion mechanism. The mean value of promotion indicates that 31% of the officials in the positions are promoted. Table 1 shows that the standard variance of all samples is greater than the mean, which indicates that the samples are sufficiently diversified to avoid the homogeneity of selected samples and overcome the problem of regression bias.

### 3.2. Effect of Water Quality Monitoring on Health

Table 2 reports the regression results of the RDD model (1), where each column is a separate regression model. With Sick1 (illness in the past four weeks) as the explained variable and the river water quality monitoring station as the cutoff, the function $f\left({\mathrm{distance}}_{it}\right)$ in the model was used in the three function forms, triangular, Epanechnikow, and uniform kernel equations for regression. Based on Table 2, the regression coefficients (treatment effect) were significantly negative at the 5% and 10% confidence levels. The regression results implied that residents located upstream of the river were less likely (by a factor of ~0.1) to have been ill in the past four weeks than their counterparts downstream. Another regression with different populations and GDP is reported in Appendix A, which shows that our regression design in Table 2 is robust.

By using the first column of Table 2 as an example, a graph of the RDD was drawn (Figure 3). The dotted line in the figure represents the critical point, indicating the location of the river water quality monitoring station. The left side of the critical point is downstream of the river and the right side is upstream. Based on comparing the fitting conditions of the control group on the left of the critical point and the experimental group on the right, the fitting line has a cutoff with a jump down at the critical point.

### 3.3. Robust Analysis of Placebo Test

To test the robustness of the RDD, the placebo test was conducted 500 times, using randomly false river water monitoring stations as cutoffs in the RDD [19,20]. Figure 4 depicts the distribution of the coefficients obtained from the 500 regressions. The results show that the coefficients estimated based on random samples are distributed around 0 (the coefficient distribution is not significantly different from 0) and the coefficients of Table 2 (red dash dotted line) are outside the range of the distribution that was obtained from the regressions of the false monitoring station samples. In other words, the treatment effects of false water quality monitoring stations do not correlate with residents’ health. This indicates that the causal analysis of the water quality monitoring station is not disturbed by the omitted variables.

### 3.4. Robust Analysis of Alternative Health Measures

Table 3 reports the alternative health variables in RDD and reports the behavior and cost of medical treatment. Columns 1 and 2 of Table 3 show that the upstream residents were less likely (by a factor of 0.0218 and 0.0278, respectively) to seek medical care or consult rural doctors in the past four weeks compared with their counterparts downstream. The absolute value of the coefficient in column 1 in Table 3 is smaller than in column 2. The last column of Table 3 shows that the upstream residents’ expenditure with respect to health decreased by ~93.31 Yuan in the past year compared with residents located downstream.

### 3.5. Robustness Analysis of Spatial Heterogeneity

With an increase in the spatial distance, pollutants are also diffused and diluted [21]. Therefore, the buffer zone of the water quality monitoring station was limited to 8, 6, and 4 km. Then, we ran the RDD with each sample and the results are presented in Table 4. The coefficients of the regression discontinuity results are all significantly negative, which aligns with the research expectations. Note that the absolute value of the coefficient increases with decreasing buffer radius. This indicates that the closer the distance to the monitoring station is, the stronger is the causal relationship between the cutoff and residents’ health. Simultaneously, the RDD with different buffer samples verifies the robustness of the bandwidth selection.

### 3.6. Robustness Test to Exclude the Accumulation Effect of River Pollution

Although river pollutants are diluted by the flow of water [21], the accumulation of pollutants downstream of the river when upstream pollutants spread downstream may affect the upstream–downstream disparity [22]. Therefore, we used the pollutant emissions of enterprises upstream of the river as the explanatory variable and performed regression with the explanatory variable of the health of residents downstream of the river, which tested the robustness of the RDD. As shown in Table 5, all regressions were statistically insignificant and the signs of the regression coefficients were not uniformly positive, indicating that upstream pollution had no statistically significant effect on the downstream residents’ health, medical treatment behavior, and medical expenditure. Therefore, there is no evidence that upstream pollution flows downstream and affects the health of downstream residents in this study.

### 3.7. Effects of Official Promotion

In this study, the DID model was used to identify the mechanism effect of officials’ promotion on pollution emissions. Table 6 shows the impact of the local officials’ promotion on industrial wastewater emissions, ammonia–nitrogen emissions, and chemical oxygen demand. The regression coefficients are statistically significant, and the signs align with the expectations. Three conclusions can be drawn from the first column in Table 6: first, the emissions of industrial wastewater in the upstream area are 80.72% ($e^{-1.6459}-1=0.8072$) lower than those in the downstream area; second, compared with unpromoted officials, promoted officials reduce the discharge of river industrial wastewater by 19.61% ($e^{-0.2183}-1=-0.1961$); third, the promotion of officials leads to a 75.39% reduction in industrial wastewater emissions in the upstream region compared with the downstream area ($e^{-1.4020}-1=-0.7539$). Column (2) and Column (3) lead to the same conclusions. Moreover, the robustness test of different regional characteristics is shown in Appendix B.

### 3.8. Effects of Pollutant Emissions

To verify the correlation between upstream–downstream disparity and the residents’ health, the DID model was used to perform a quantitative analysis. Table 7 shows that the main effect of the pollution variable is significantly positive, which indicates that the probability of residents becoming sick in the past four weeks increased with the increase in pollutant emissions from firms. The main effect of the upstream variable is significantly negative, indicating that residents living upstream were also less likely to have been ill than those living downstream. Finally, the coefficient of the interaction term of the pollution variable and upstream variable is significantly negative, which indicates that upstream pollution discharge reduced the probability of residents becoming sick compared with their counterparts downstream. An additional estimation of different regional characteristics is reported in Appendix C.

## 4. Discussion

In this study, river water quality monitoring stations were used as a geographic cutoff to divide the study samples into a treatment group upstream and a control group downstream. This mechanism was used to conduct a local random quasi-natural experiment. Using the RDD, DID, and OLS approaches, this study found that upstream pollution (industrial wastewater emissions, ammonia nitrogen emissions, and chemical oxygen demand production) of river water quality monitoring stations is significantly lower than that downstream, and the prevalence rate of residents upstream of the river in the past four weeks was lower than that of residents downstream. The robustness tests show that residents upstream and downstream of the river exhibit different medical treatment behaviors and expenditures. In addition, the health differences between upstream and downstream residents are caused by the promotion of government officials.

Most studies showed that pollution in downstream rivers is more severe than in upstream rivers [12,23,24,25,26]. The results of this study further reveal the reasons behind this disparity in pollution. Local officials in China adopt biased environmental pollution control strategies under the economic and environmental promotion incentive mechanisms. They are more concerned about controlling upstream pollution, which affects pollution monitoring data and relaxes the supervision of downstream firms. In addition, the correlation between the promotion of local officials and the upstream and downstream pollution gap, and the effect of this correlation on the residents’ health, were analyzed in this study. The effects of this pollution phenomenon on the illness of residents in the past four weeks and the residents’ medical treatment behavior were quantified. In addition, the samples selected for this study included almost all rivers in the region, and evidence was obtained for a wider range rather than a few rivers.

This study is unique in that we aimed to evaluate the water regulations, identify possible loopholes in China’s officials’ promotion incentive policy for water protection, and estimate its effects on residents’ health, which strongly contributes to the control of the water environment and the protection of residents’ health. In contrast to previous studies [27,28,29,30,31], our research was based on the scientific quasi-natural experiment regression method and used more detailed data according to the location of the water quality monitor station, the locations of residents and their health status, and the locations of rivers.

First, the rich data sets allowed us to obtain a causal inference and identify a generic relationship between an effect and the cause; thus, the effect and cause could not be mixed. We collected data sets for 1907 river water quality monitoring stations on 1007 rivers and biographical information for 5032 officials in 2606 counties. Furthermore, the standard variance of all samples was greater than the mean, as Table 1 shows, which indicates that the samples were sufficiently diversified to avoid the homogeneity of selected samples and overcome the problem of regression bias.

Second, the effect of river pollution on the residents’ health was quantified based on a quasi-natural experiment of a geographic regression discontinuity design. The regression discontinuity design (RDD) is a quasi-experimental pretest–posttest design that aims to determine the causal effects of interventions by assigning a cutoff or threshold above or below which an intervention is assigned. The water monitoring station was exactly this cutoff in the study. Comparing observations lying closely on either side of the threshold makes it possible to estimate the average treatment effect in environments.

Third, the evidence of health disparities caused by upstream and downstream pollution differences was provided based on a quasi-natural experiment and a new influencing mechanism was identified. Local government officials’ promotion behavior in pursuit of environmental performance increases the disparity in pollution between upstream and downstream areas. This agrees with discussions in the literature on how officials’ promotion behavior affects environmental pollution [12,32,33,34]. However, we further discussed the effects of this behavior on residents’ health and medical behavior. The results showed that such behavior increases the gap in residents’ health.

Moreover, we conducted a series of robustness tests to ensure that the regressions were reliable. First, we obtained a zero mean parameter distribution from the 500-times placebo regression under virtual random samples of river water quality monitoring stations, which indicated that the health effect of the virtual river monitoring station was zero. Second, the alternative explained variables that measured the residents’ health were also robust. Local officials’ pursuit of environmental promotion in China has led to a health gap between upstream and downstream residents, which has affected residents’ healthcare practices and spending. Finally, the regression of upstream river pollution on downstream residents’ health excluded the cumulative effects of river pollution, which showed that the correlation between them was weak and that the cumulative effect of pollution did not interfere with the correlation between government officials’ behavior and residents’ health.

The effect of digital technology on the correlation between local government officials’ behavior and residents’ health is a possible topic for future research. The development of information technology and real-time dynamic monitoring can help the central government to supervise local officials’ progress in achieving environmental protection goals. This may help to narrow the gap between the central government’s environmental target and the local government’s implementation effect. In addition, applying digital technologies in environmental governance also helps the public to obtain more reliable pollution information and implement pollution prevention measures, thus significantly improving social welfare.

This study has several limitations. Although we quantified how the promotion behavior of local government officials affects residents’ health based on a natural experiment, only nine of 32 provinces’ health data in China were included in our study due to data limitations, which prevented an analysis of China’s national environmental governance situation. However, the population and GDP of the nine provinces surveyed by the CHNS account for 41.49% and 40.84% of China’s population and GDP, respectively, and there are 39.91% of river water quality monitoring stations located in these nine provinces, confirming the representativeness of the results of this study. Moreover, despite the outdated health data, the unchanged water regulation policy and upstream–downstream pollution gap guaranteed robust results.

## 5. Conclusions

In this study, we used river pollution control as an example to assess how the pursuit of promotions by local officials in China affects the health of residents and to determine the mediating influence mechanisms of pollutant emissions. Allowing the local government to authorize river water regulations may cause principal–agent problems because incentivizing promotions within public departments may lead to differences in environmental monitoring. Although local officials have the motivation to pursue better environmental governance, their pursuit of promotions may lead to behavior that negatively affects residents’ health. The empirical study shows a causal relationship between the upstream–downstream disparity and residents’ health caused by the promotion behavior of officials in pursuit of environmental performance. Key results/findings are summarized as follows:

(1) China’s rapid economic development and industrialization since the 1970s have led to a sharp deterioration in water quality. Since early 2000, China has implemented strict water pollution control measures, including a target-based system to improve water quality. The central government has set up almost two thousand water monitoring stations along the river and used the water quality readings as a critical criterion for local government officials’ promotion.

(2) Readings from water quality monitoring stations can only reflect the water quality upstream of rivers, giving local government officials the promotion incentive to supervise polluting enterprises located upstream while relaxing regulations on polluters downstream. We used this spatial disparity to quantify the effects of watering down environmental regulation on residents’ health.

(3) The quasi-natural experiments suggest that local officials more intensively supervise upstream polluting firms, with residents living upstream suffering from less water pollution than their downstream counterparts, and thus being 10.59 percent less likely to be sick in the past four weeks.

(4) The empirical estimation found that environmental regulation led to significantly higher medical costs for downstream residents. The medical expenditure of downstream residents is 93.31 Yuan more than that of upstream residents. Taken together, China’s efforts in reducing water pollution led to a total loss in medical expenditure of 462.75 billion Yuan from 2000 to 2011.

(5) Further, the study provides a timely assessment of the central government’s environmental target-oriented policy. It demonstrates an apparent misalignment between the central government’s environmental goal and the local government’s official promotion incentive, as the local official prioritizes monitoring station readings over actual river water quality, conflicting with the intentions set by the policy.

## Figures and Tables

**Figure 1 ijerph-19-16770-f001:**
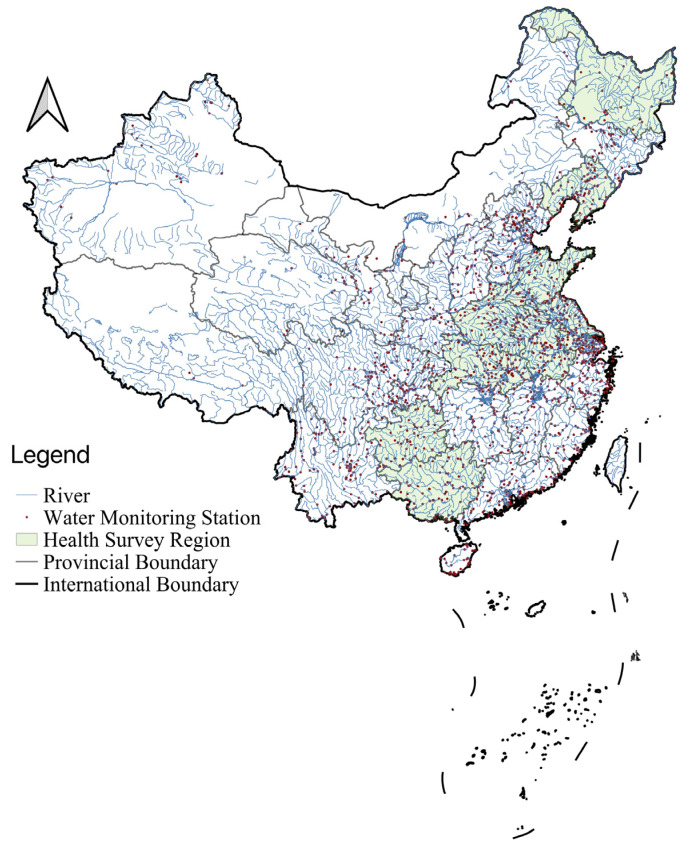
Distribution of main river water quality monitoring stations and health survey regions in China.

**Figure 2 ijerph-19-16770-f002:**
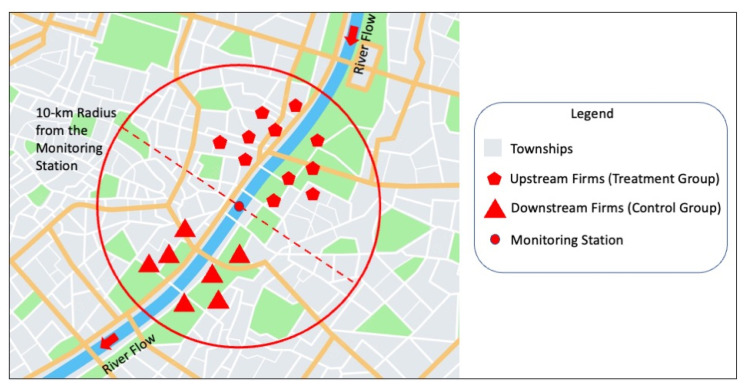
Diagram of regression discontinuity design strategy.

**Figure 3 ijerph-19-16770-f003:**
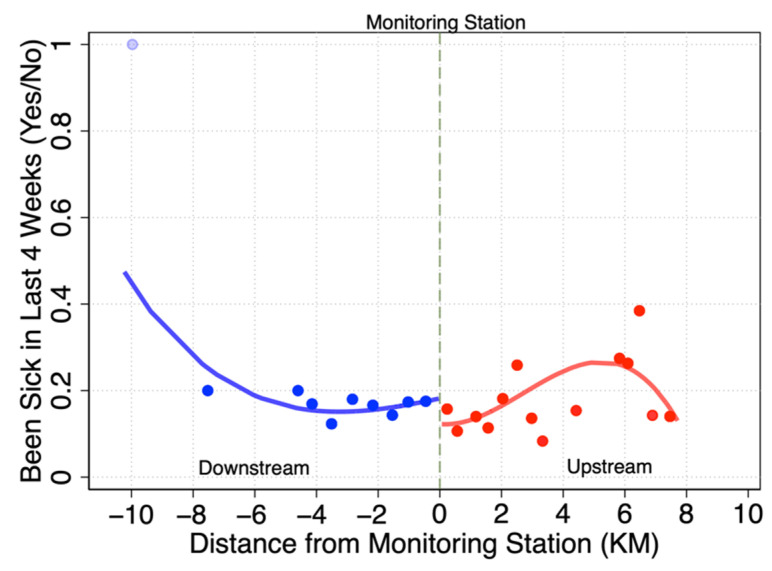
RDD plot of the effects of water quality monitoring on health.

**Figure 4 ijerph-19-16770-f004:**
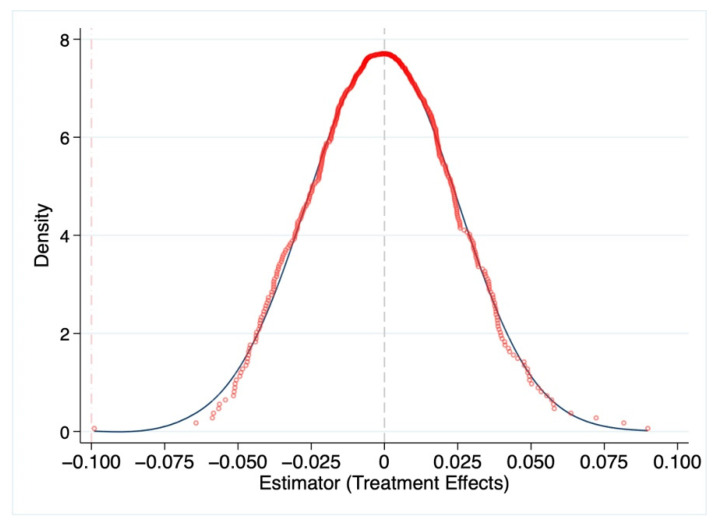
RDD placebo test.

**Table 1 ijerph-19-16770-t001:** Descriptive statistical results for the main variables.

Variable		Description	Obs.	Mean	Std. dev.	Min	Max
Dependent Variables	Sick1	been sick in the last 4 weeks (1 for sick and 0 for healthy)	7088	0.16	0.37	0	1
	Sick2	sought formal medical care in the last 4 weeks (1 for yes and 0 for no)	4863	0.02	0.15	0	1
	Sick3	visited a folk doctor in the last 4 weeks (1 for yes and 0 for no)	5624	0.04	0.20	0	1
	Sick4	money spent on illness (Yuan)	1517	129.92	254.58	0	1000
Driving Variable	Distance	distance between the enterprise and the monitoring site (KM)	7640	−0.38	2.55	−9.98	9.71
Interaction Variables 1	Pollution1	wastewater emissions (Tons)	5018	34.39	544.83	0	33,423.60
	Pollution2	ammonia nitrogen emission emissions (Tons)	3833	70.24	575.75	0	18,416.18
	Pollution3	chemical oxygen demand production (Tons)	7316	107.96	424.46	0	6971.69
Interaction Variables 2	Promotion	official promotion (1 for promotion and 0 otherwise)	8955	0.31	0.46	0	1

Abbreviations: Obs., observations; Std. dev., standard deviation; min, minimum; max, maximum.

**Table 2 ijerph-19-16770-t002:** Upstream–downstream health gap.

	(1)	(2)
Variables (Kernel Type)	Sick1 (Triangular)	Sick1 (Uniform)
Treatment Effect	−0.1059 **	−0.1147 **
	(0.0539)	(0.0528)
Observations	7092	7092

Standard errors are given in parentheses. ** *p* < 0.05.

**Table 3 ijerph-19-16770-t003:** Robustness of alternative dependent variables.

	(1) Formal Medical Treatment	(2) Folk Doctor Treatment	(3) Expenditure on Illness
Variables	Sick2	Sick3	Sick4
Treatment Effect	−0.0218 **	−0.0279 **	−93.3055 **
	(0.0088)	(0.0157)	(45.7887)
Observations	4853	5629	1310

Standard errors are given in parentheses. ** *p* < 0.05.

**Table 4 ijerph-19-16770-t004:** Robustness of different buffer distances.

	(1)	(2)	(3)	(4)
	Buffer < 10 km	Buffer < 9 km	Buffer < 8 km	Buffer < 7 km
Variables	Sick1	Sick1	Sick1	Sick1
Treatment Effect	−0.1061 **	−0.1070 **	−0.1091 **	−0.1270 **
	(0.0526)	(0.0541)	(0.0547)	(0.0828)
Observations	7090	6853	6744	6623

Standard errors are given in parentheses. Observations (people). ** *p* < 0.05.

**Table 5 ijerph-19-16770-t005:** Effects of upstream pollutant emissions on downstream residents’ health.

	(1)	(2)	(3)	(4)
	Been Sick	Formal Medical Treatment	Folk Doctor Treatment	Expenditure on Illness
Variables	Sick1	Sick2	Sick3	Sick4
Pollution1	0.0226	0.0129	−0.0009	15.7721
	(0.026)	(0.0115)	(0.0081)	(27.1844)
Pollution2	−0.0138	0.0069	−0.0034	−2.9186
	(0.011)	(0.0045)	(0.0034)	(26.7925)
Pollution3	0.0309	−0.0024	0.0040	2.3232
	(0.024)	(0.0104)	(0.0074)	(16.5924)
Observations	6433	5824	6432	6245
R-squared	0.5441	0.4651	0.3870	0.4892

Standard errors are given in parentheses.

**Table 6 ijerph-19-16770-t006:** Effect of officials’ promotion on pollution emissions.

	(1)	(2)	(3)
	Industrial Wastewater Emissions	Ammonia–Nitrogen Emissions	Chemical Oxygen Demand
Variables	Pollution1	Pollution1	Pollution1
Promotion	−0.2183 **	−1.8497 ***	−1.6777 ***
	(0.1068)	(0.2193)	(0.0007)
Upstream	−1.6459 ***	−3.7965 ***	−1.6634 ***
	(0.1380)	(0.3051)	(0.0008)
Upstream × Promotion	−1.4020 ***	−2.3831 ***	−2.2328 ***
	(0.1110)	(0.2465)	(0.0007)
Observations	5526	5680	6128
R-squared	0.6382	0.7107	0.6390
City FE	Yes	Yes	Yes
Year FE	Yes	Yes	Yes

Standard errors are given in parentheses. *** *p* < 0.01, ** *p* < 0.05.

**Table 7 ijerph-19-16770-t007:** Effect of pollutant emissions on residents’ health.

	(1)	(2)	(3)
Variables	Sick1	Sick1	Sick1
Upstream	−0.0193 **	−0.0212 ***	−0.1932 **
	(0.0839)	(0.0003)	(0.0839)
Pollution1	0.0356 ***		
	(0.0129)		
Pollution1 × Upstream	−0.0375 **		
	(0.0189)		
Pollution2		0.0009 ***	
		(0.0002)	
Pollution2 × Upstream		−0.0135 ***	
		(0.0003)	
Pollution3			0.0356 ***
			(0.0129)
Pollution3 × Upstream			−0.0375 **
			(0.0189)
Observations	1580	1580	1580
R-squared	0.466	0.400	0.408
City FE	Yes	Yes	Yes
Year FE	Yes	Yes	Yes

Standard errors are given in parentheses. *** *p* < 0.01, ** *p* < 0.05.

## Data Availability

The data for this study are available from the authors upon request.

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
