# Peer review of "Effect of Watering down Environmental Regulation on Residents’ Health in China: A Quasi-Natural Experiment of Local Officials’ Promotion Motivation"

_ijerph, 2022, doi:10.3390/ijerph192416770_

Round 1

Reviewer 1 Report

This study divides the upstream and downstream river samples by river water quality monitoring stations, uses regression analysis to analyze the health difference between residents in the upstream and downstream of river water quality monitoring points, and quantitatively evaluates the impact of local government officials' pursuit of environmental performance promotion behavior on residents' health. From the promotion mechanism of local officials to human health, propose possible loopholes in the mechanism, which has a strong contribution to the control of water environment. the analysis method of this paper is scientific and rigorous with good logic. However, some data presentation and paragraph writing can still be improved:

1. The high proportion of provincial population and GDP does not have the ability to represent China's basic national conditions. It is suggested to select regional data of other provinces for analysis.

2. When analyzing the impact of promotion motivation of local officials on the health of residents in the jurisdiction area, it is suggested to make a more detailed analysis by distinguishing the characteristics of differences in the number of patients, the distance between monitoring stations, pollutant emissions, and the number of promoted people in different regions.

3. It is recommended to add the downstream pollutant data acquisition method in the data source and variables part.

4. The data subdivision visualization in this paper needs to be improved. Although each table gives detailed data information, the effect is poor. It is suggested to refine the data in this part to make the paper more clear.

5. The language still needs improvement. It is suggested that the discussion paragraph be further modified and condensed.

6. The layout of some paragraphs needs to be improved.

7. It is recommended to quote these articles: [1] Jing Xu, Guangqiu Jin, Hongwu Tang, Yuming Mo, You-Gan Wang, Ling Li. Response of water quality to land use and sewage outfalls in different seasons[J]. Science of the Total Environment,2019,696. [2] Na Wang, You-Gan Wang, Shuwen Hu, Zhi-Hua Hu, Jing Xu, Hongwu Tang, Guangqiu Jin. Robust Regression with Data-Dependent ​Regularization Parameters and Autoregressive Temporal Correlations [J]. Environmental Modeling & Assessment, 2018, 23(6): 779-786.

Reviewer 2 Report

Due to many models employed at once (do all of them are needed indeed to conclude then that officials are guilty in the water quality and public health?) and not very clear presentation of the results, manuscript is difficult to read and understand. It needs more clear presentation throughout the text, it can be shortened too.

Reviewer 3 Report

Title: “Effect of watering down environmental regulation on residents' health in China: A quasi-natural experiment of local officials' promotion motivation”.

After reviewing the present manuscript, reviewer found that the authors made interesting work and noticed that this manuscript is fit with MDPI-IJERPH and has following general comments and specific comments with regards to the improvement of the manuscript prior to the publication.

The manuscript is written in good English despite some minor errors. The content of the manuscript is well organized, and the length and depth of the work is fair and sufficient for an article.

Be consistent with the space between units and numbers. Use of stops in appropriate places and correct way of referencing must be carefully looked.

The comments below are organized in order as they appear in the manuscript.

1.                 Graphical abstract -Authors are recommended to include a graphical abstract.

2.                 It is suggested for the authors to include the highlights, to present the most significant findings of the research.

3.                 Keywords-Each keyword should start with an upper case.

4.                 Line 66-Correct the sentence “The aims of this study were to quantify how the difficulties”

5.                 Figure1- Authors are suggested to improve the figure quality with clear label’s

6.                 Line134-Change of font was observed, correct it

7.                 References-Follow journal guidelines

Round 2

Reviewer 1 Report

This study divides the upstream and downstream river samples by river water quality monitoring stations, uses regression analysis to analyze the health difference between residents in the upstream and downstream of river water quality monitoring points, and quantitatively evaluates the impact of local government officials' pursuit of environmental performance promotion behavior on residents' health. From the promotion mechanism of local officials to human health, propose possible loopholes in the mechanism, which has a strong contribution to the control of water environment. the analysis method of this paper is scientific and rigorous with good logic.I hope the author has further research results.

Round 3

Reviewer 2 Report

Lines 91-92, you wrote: 1) was used to determine the social and economic status, health services to their dietary preferences – probably, should be: “and their dietary preferences” or so?

Lines 100-124 are missed in the provided file and I can’t check

Line 133, you wrote: “First, A script was written”. Should be: “First, a script was written”

Lines 136-138: What firms are mentioned here? Enterprises? Companies? It’s better to use some synonym. By the way, in response-to-reviewer file you wrote that there were “sewage firms”. So, sewage should be added here too. Also, what higher elevation is mentioned here? The one above a sea level, right?

Line 190-192 of the revised version. You wrote: Interaction is a variable that uses pollution emission data when test pollution mechanism or use official promotion when test promotion mechanism; official promotion data? Official promotion of who? Maybe data of officials’ promotion?

Line 198 corresponds with the Response 16. In the response you wrote a new expression as follows: “the results of which are reported in the next section” and this is correct in terms of grammar. However, in the text of a revised Manuscript this point looks like “…results were reported in the next section”, which is incorrect. Replace it by the correct version, please.

Table 1. Shift KM and Tons, not Ton to the third column as it was done for the rest of units here

Meanwhile, here and under the rest of tables, change expression “Standard errors in parentheses” to “Standard errors are given in parentheses, **p < 0.05” or so, or “Values in parentheses represent standard errors”.

Response 21, you did nothing. The phrase: “This means that there is a causal relationship between upstream-downstream disparity and the residents' health caused by the promotion behaviour of officials in pursuit of environmental performance” is on its place now. It’s not good to lie.

Response 25 is not true again. You did nothing with it. The sentence mentioned is as follows: “Therefore, we believe that the absolute values of the coefficients in columns 1 and 2 in Table 3 are smaller than those in column 2”. Using simple logic the one can understand that it is illogical for absolute values of the coefficients in columns 2 to be smaller than those in column 2. So, this sentences needs correction. Do it, please. These are lines 271-273 now.

Table 4 – insert units of observations for people into the table by itself, e.g. “Observations (people)”, remove p<0.1 as these data seem not to be present in the Table now.

Response 34 seem to be good for the Discussion, as it is explaining well the relation between variables. But I haven’t found this paragraphs in the text. They could be incorporated into the Discussion.

One more question: in your work you use both data of self-assessment of people asking if they have been sick in last 4 weeks. Another bulk of the health data is “The China Health and Nutrition Survey (CHNS) carried out by the Chinese Centre for Disease Control and Prevention and the American University of North Carolina during the period 1989–2011. How these old data reflect the modern state. Are you sure that it can be used after almost 12 years after it was obtained and can be combined with the present health information?
